# Compression theory for inhomogeneous systems

Doruk Efe Gökmen ⬡ 1,2,3,4 ✉, Sounak Biswas[5], Sebastian D. Huber ⬡ [1], Zohar Ringel[6], Felix Flicker[7] ✉ & Maciej Koch-Janusz ⬡ 2,8,9 ✉

The physics of complex systems stands to greatly benefit from the qualitative changes in data availability and advances in data-driven computational methods. Many of these systems can be represented by interacting degrees of freedom on inhomogeneous graphs. However, the lack of translational invariance presents a fundamental challenge to theoretical tools, such as the renormalization group, which were so successful in characterizing the universal physical behaviour in critical phenomena. Here we show that compression theory allows the extraction of relevant degrees of freedom in arbitrary geometries, and the development of efficient numerical tools to build an effective theory from data. We demonstrate our method by applying it to a strongly correlated system on an Ammann-Beenker quasicrystal, where it discovers an exotic critical point with broken conformal symmetry. We also apply it to an antiferromagnetic system on non-bipartite random graphs, where any periodicity is absent.

Dramatic improvements in data availability, stemming from both experiments and simulation, have enabled the exploration of increasingly complex physical systems. A glut of raw data does not, however, equate understanding, particularly when its processing easily exceeds our computational resources. A key objective is to distil data into a succinct theory in terms of appropriate collective variables that uncover and summarise the essence of the system. Renormalization Group (RG) approaches in statistical physics provide a systematic path towards that goal[1,2]. However, both identifying the relevant degrees of freedom (DOFs), as well as executing the mathematical procedure deriving the effective theory[3,4] are often very challenging in inhomogeneous systems when prior intuition is scarce.

Many complex systems are inhomogeneous, owing their properties precisely to the lack of translational invariance. Problems as disparate as biological tissue mechanics[5–7] and properties of metallic glasses[8,9] can be cast as statistical mechanical problems on irregular graphs. The ultimate goal would be a generic understanding of the emergent properties of such systems, much as RG provided an understanding of critical phenomena in translation-invariant systems, where the proxy of wavelength can be used to organise the modes to target the relevant low-energy operators[10]. While one can still implement scale transformations to perform RG for inhomogeneous systems, the lack of a clear proxy like wavelength makes it necessary to carefully craft the coarse graining locally, particularly since real-space RG can be ill-defined under certain poor choices[11].

Here we tackle the challenge of inhomogeneity in complex systems with vast configuration spaces. Formulating the RG of an inhomogeneous system as a lossy compression of information[12,13] on a graph allows us to define the procedure in a geometry-independent manner. This key step is based on the observation that the compression theoretic RG some of us introduced for lattice systems[14–16] can be mathematically generalised to arbitrary graphs, yielding RG informed both of the interactions and of the spatial relations, thus overcoming a major conceptual challenge. The numerical execution of this data-driven procedure entails the difficult task of estimating mutual information for large-dimensional random variables on graphs. This is

¹Institute for Theoretical Physics, ETH Zurich, Zurich, Switzerland. ²James Franck Institute, The University of Chicago, Chicago, IL, USA. ³Department of Statistics, The University of Chicago, Chicago, IL, USA. ⁴National Institute for Theory and Mathematics in Biology, Chicago, IL, USA. ⁵Institut für Theoretische Physik und Astrophysik, Universität Würzburg, Würzburg, Germany. ⁶Racah Institute of Physics, Hebrew University, Jerusalem, Israel. ⁷School of Physics, Tyndall Avenue, Bristol BS8 1TL, UK. ⁸Department of Physics, University of Zürich, Zürich, Switzerland. ⁹Haiqu, Inc., San Francisco, CA, USA.
✉ e-mail: gokmen@uchicago.edu; flicker@physics.org; maciej@haiqu.ai

achieved by using recent advances in machine learning, formulating it as a classification task of distinguishing jointly sampled pairs of variables from those sampled independently (see 'Methods'), i.e. using contrastive learning[17–19], and extending the computational tools of refs. 20,21.

Our algorithm assumes the system is defined on a graph $G$ and explicitly constructs new effective DOFs $\mathcal{H}^i$ from local configurations of DOFs $\mathcal{V}^i$ supported on vertices or edges of local subgraphs $V^i$ of $G$. This is achieved by a coarse graining transformation, which we parametrise as a linear neural network (NN) with a set of parameters $\mathbf{\Lambda}^i$ and a non-linear discretisation map $\tau$:

$$\mathcal{H}^i = \tau\left(\mathbf{\Lambda}^i \cdot \mathcal{V}^i\right). \tag{1}$$

This coarse-graining is optimised locally in region $i$ to maximise the mutual information $I(\mathcal{H}^i : \mathcal{E}^i)$ with the configurations supported on the spatial environment $E^i$ of $V^i$ (see Fig. 1 and 'Methods')[14–16]:

$$\mathbf{\Lambda}^i := \arg\max I(\mathcal{H}^i : \mathcal{E}^i). \tag{2}$$

The disjoint spatial environment subgraph $E^i$ is defined using the graph distance (see Fig. 1 and Fig. S1), and its separation from the block $V^i$ has a crucial role of screening out the irrelevant short-range correlations. The mutual information

$$I(\mathcal{X} : \mathcal{Y}) = H(\mathcal{X}) + H(\mathcal{Y}) - H(\mathcal{X}, \mathcal{Y}), \tag{3}$$

quantifies the amount of information the random variable $\mathcal{X}$ reveals about the other $\mathcal{Y}$, by measuring their overlapping contributions to the total entropy $H$.

The maximisation is performed under the constraint that the coarse-grained variable $\mathcal{H}^i$ is subject to a discretisation $\tau$ ('Methods').

This crucial aspect of the variational principle renders the coarse-graining a *lossy* compression map, i.e. $I(\mathcal{H}^i : \mathcal{E}^i) < H(\mathcal{V}^i)$, where $H$ is the entropy. Thus, only the most relevant collective features of $\mathcal{V}^i$ that survive this information bottleneck will be stored in $\mathcal{H}^i$. Note that they depend on both the topology of the graph $V^i$ and the interactions of the DOFs $\mathcal{V}^i$ on it, and information about both is contained in the statistics of the samples $(\mathcal{V}^i, \mathcal{E}^i)$.

The variational principle in Eq. (2) provides a powerful substitute for heuristic approaches. Specifically, rather than guessing an important local collective property for coarse graining, the collective DOFs are instead designed by the statistics of their environments. This is essential for moving beyond translation invariance. While the coarse graining $\mathbf{\Lambda}^i$ erases microscopic fluctuations, its local optimisation allows it to retain the distinct qualitative characteristics that emerge in different spatial regions across an inhomogeneous system. This can even be reflected in a non-uniform cardinality of compressed variables $\mathcal{H}^i$, as illustrated in Fig. 1. Thanks to the recently discovered links between RG relevance and compression theory[13], this procedure eliminates proxies like wavelength (or even energy, in purely entropic systems) and grants direct access to the operators with low-scaling dimension[14].

We illustrate the power of this approach by explicitly constructing emergent DOFs to confirm an open conjecture regarding the presence of discrete scale invariance (DSI)[22–24] of a strongly correlated statistical model on quasicrystals[25]. This problem is perfectly suited to our method: the combination of constraints and aperiodicity provides a serious challenge for human intuition[25–27]; yet efficient algorithms exist that can generate huge data sets for training machine intuition[28].

Though quasicrystals lack translational invariance, they possess long-range order[29,30]. To show the generality of our method, and its independence of the existence of any special tilings, we also applied it to frustrated antiferromagnets on non-bipartite random graphs lacking any (quasi)periodicity, where it finds the optimal bipartitioning

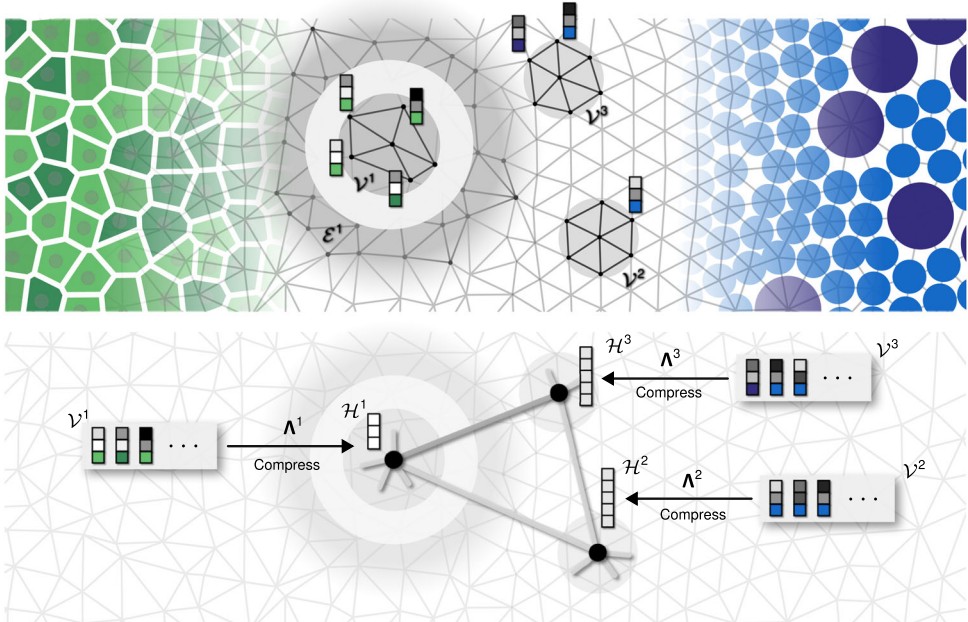

**Fig. 1 | Schematic for constructing collective degrees of freedom in inhomogeneous systems.** Distinct systems like tissues (left, in green) and colloidal suspensions (right, in blue) can be abstracted into a set of vector degrees of freedom $\mathcal{V}^i$ (indicated by stacks of squares, $i = 1, 2, 3$) living on an irregular graph with local structure. The final component of each vector is shown by a coloured box to indicate potentially different types of internal degree of freedom, unique to each sub-system. To derive a compressed representation of such systems, it is essential to tailor the coarse graining transformation $\mathbf{\Lambda}^i$ for each local neighbourhood $i$. This is achieved by an information theoretic variational principle, where $\mathbf{\Lambda}^i : \mathcal{V}^i \mapsto \mathcal{H}^i$ maximises the mutual information $I\left(\mathcal{H}^i : \mathcal{E}^i\right)$. This allows the compressed variables $\mathcal{H}^i$ to capture the emergent long-range physics according to the statistics of the surrounding distant environment $\mathcal{E}^i$. Local optimisation can produce compressed variables with varying cardinality across the system, here illustrated by vectors $\mathcal{H}^i$ with varying numbers of components. The connectivity of the emergent *super-graph* is determined through the correlations of the new variables.

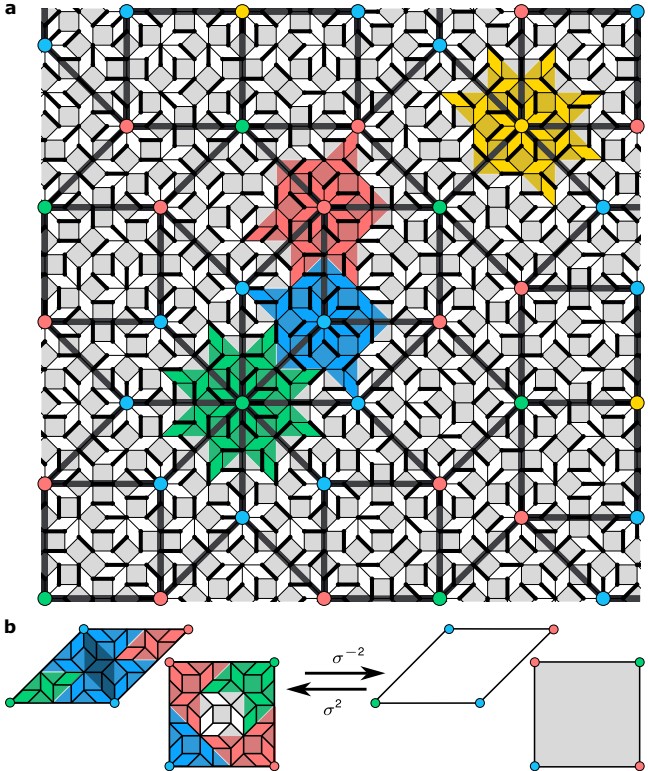

**Fig. 2 | Self-similarity of the Ammann-Beenker tiling, and the coarse graining blocks. a** A microscopic dimer configuration on the AB tiling's edges, with an overlaid AB *super-quasilattice*, self-similar to the microscopic one. The effective degree of freedom at a supervertex with valence $n$ will be obtained by coarse graining the dimer configuration in the surrounding polygon tile $V^n$. In total there are 4 classes of such polygons, here shown in green, blue, red and yellow for $n = 8, 3, 4, 5$, respectively. The shape of the block tile is dictated by the valence $n$ of the central supervertex in matching colour. **b** The inflation (deflation) $\sigma^{2(-2)}$ of the elementary rhombi and squares generating the tiling, with parts of the polygonal domains indicated in colour. Coarse graining all such polygonal patches executes a deflation $\sigma^{-2}$ of the original AB quasilattice, yielding the super-quasilattice shown.

(Section IV of Supplementary Information). Our graph-based tool leverages the computational back-end of the real-space mutual information neural estimation (RSMI-NE) package[20,21], allowing to efficiently explore large-scale phenomena.

## Results

### Dimers on the Ammann-Beenker tiling

The Ammann-Beenker (AB) construction gives quasiperiodic tilings of the plane using two distinct plaquettes: a rhombus and a square[29–31]. Like their famous cousins, the Penrose tilings[32], AB tilings feature diffraction patterns exhibiting crystallographically 'forbidden' symmetries, here 8-fold[29]. Likewise, they can also be generated by a recursive procedure in which an *inflation* map $\sigma$ acts on a small seed patch by composing the constituent plaquettes as shown in Fig. 2b, and subsequently rescaling edge lengths by the silver ratio $\delta = 1 + \sqrt{2}$. A special role is played by 8-fold coordinated vertices: under inflations all lower coordinated vertices ultimately become (and stay) 8-vertices. Each 8-vertex is characterised by an *order*, i.e. the maximal number of inverse *deflations* $\sigma^{-1}$ after which it still remains 8-fold coordinated. The order of an 8-vertex intuitively specifies the maximal size of the local 8-fold symmetric patch centred on it. The quasiperiodic AB lattice ('quasilattice') is thus invariant under discrete rescalings. Such discrete rescalings are easily visualised for even order deflations $\sigma^{-2n}$ by drawing a super-quasilattice connecting 8-fold vertices (Fig. 2a).

In this setting, we consider a dimer model. Dimer models enjoy a deceptively simple definition: microscopic DOFs live on the links of a graph (here, the edges of the quasilattice), which can be either occupied or empty. The key element is a hard local constraint: at every vertex where the links meet, one and only one of the links is occupied by a dimer. This gives rise to a surprisingly rich phenomenology. Dimer models on regular lattices have been studied extensively, originally due to their purported relevance to high-$T_c$ superconductivity[33]. They have since been shown to support topological order and fractionalisation[34,35] and exotic critical points[36]. The classical problem is closely related to the quantum one[37,38] and has deep connections to combinatorics[39–41] and the study of random surfaces[42,43].

Recent work has begun to explore the interplay of (strongly-correlated) dimer physics and quasiperiodicity. Particularly, AB tilings, in contrast to Penrose tilings[26], host perfectly matched dimer configurations (i.e. with a vanishing density of uncovered vertices) in the thermodynamic limit. Numerically computed dimer correlations exhibit a quasi power-law decay with a complex spatial structure[25]. Moreover, the combinatorial proof of perfect matching pointed to a hierarchy of self-similar effective matching problems at different scales.

Taken together, these facts suggest a conjecture that not only the AB tilings themselves, but crucially also the physics of the dimers on the AB tilings, exhibit DSI[25]—a potentially striking and unusual example of the relevance of quasiperiodicity for critical behaviour. A proof and a microscopic physical mechanism at the level of the dimer ensemble has, however, proven elusive. The putative criticality naturally calls for an RG analysis, but general RG approaches for quasiperiodic systems in $D \geq 2$ dimensions are in their infancy.

In the following, we will first identify the natural block structures to coarse grain the dimer DOFs with a certain scale transformation of the quasilattice. We will then use our compression approach based on Eq. (2) to address two key questions regarding the dimer model on AB tilings: What are the collective coarse-grained DOFs, and what is the structure of their correlations? Finally, by analysing the compressed data provided by our algorithm, we demonstrate the presence of DSI in the dimer model on the AB tiling.

### Collective $\mathbb{Z}_n$ clock degrees of freedom

To construct the collective DOFs, we first need to specify the block regions $\mathcal{V}$ to be coarse grained. In the AB tiling there are natural choices, set by the recursive structure of the AB quasilattice itself[44]. At each scale, the AB tiling can be covered by four 'classes' of blocks[25] $V^n$, shown in different colours in Fig. 2, each deflating to vertices of differing connectivity $n$ in the super-quasilattice. In the following, we label each class by the corresponding connectivity $n$ of the super-quasilattice. Our method does not rely on any fine-tuned choice of the block shape (see Section III in Supplementary Information).

In each different class, the algorithm identifies the collective DOF as a $\mathbb{Z}_n$ *clock variable*[45], with $n$ the connectivity, or class, of the central supervertex of $V^n$. This is revealed by a variational compression map $\boldsymbol{\Lambda}^n$ which assigns to a Monte Carlo dimer configuration $\mathcal{V}^n$ a short binary code $\mathcal{H}^n$ (Fig. 3a). The binary digits are set by applying individual components $\Lambda_k^n$ to $\mathcal{V}^n$ (itself a long bitstring of dimer occupations in the block). Each component of the vector $\boldsymbol{\Lambda}^n$ is a priori a general nonlinear map, parametrised by a NN, whose output is finally binarized.

The length of the code is not supplied, but it is inferred by sequentially increasing the number of components in $\boldsymbol{\Lambda}^n$, and training the compression of $\mathcal{V}^n$ to optimally preserve the mutual information with its environment $\mathcal{E}^n$. Crucially, the maximal retained information about $\mathcal{E}^n$ plateaus with a different optimal code-length depending on the class $\mathcal{V}^n$. Particularly, for $V^8$ (green 8-star patch in Fig. 2) the optimal number of components is four, while for $V^3$ (blue patch in Fig. 2) it is only two (Fig. 3b,f). Further, nonlinearity of $\boldsymbol{\Lambda}$ networks does not

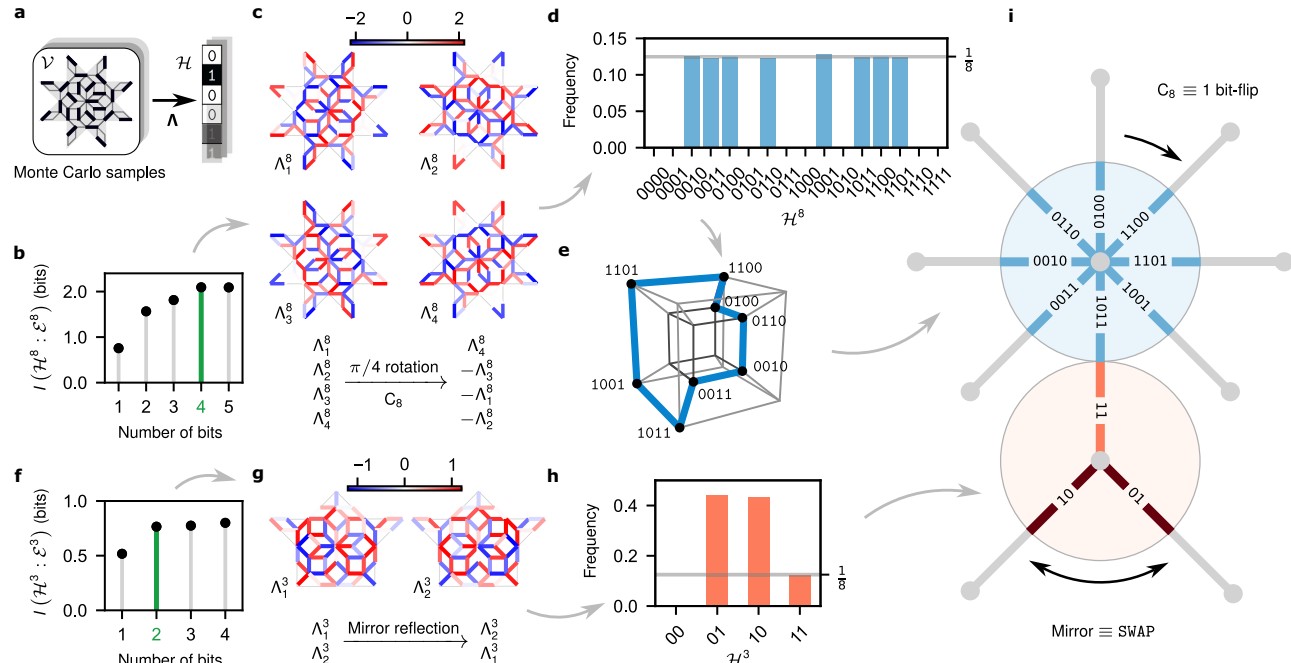

**Fig. 3 | Finding collective clock variables. a** Coarse graining transformation $\mathbf{\Lambda}$ compressing Monte Carlo configurations $\mathcal{V}$ into bitstrings $\mathcal{H}$ on supervertices of the $\sigma^{-2}$ deflated AB tiling. Each bit $\mathcal{H}_k$ is decided by the sign of the linear transformation $\Lambda_k \cdot \mathcal{V}$. **b, f** The length of the bitstring $\mathcal{H}^{8(3)}$ is determined by the saturation point (shown in green) of mutual information at 4 (2) bits at 8- (3-)supervertices. **c, g** The respective optimal filters $\mathbf{\Lambda}^8$ and $\mathbf{\Lambda}^3$ carry a representation of the local spatial symmetries of corresponding supervertices, namely $C_8$ and mirror. **d, h** The

probability distributions $P(\mathcal{H}^{8(3)})$ occupy the space of codes sparsely, and form abstract $\mathbb{Z}_{8(3)}$ clock variables. **e** In particular, $\mathcal{H}^8$ forms a closed 8-loop, where each state has exactly two neighbours with Hamming-distance 1. **i** The representations of the local symmetries on filters induce transitions between adjacent clock-states, enabling the identification of abstract clock-states with spatial directions along the links of the quasiperiodic lattice.

improve compression: the same amount of information is preserved with only linear components. Optimal linear maps on the space of dimer configurations $\mathcal{V}^n$ on subgraph $V^n$ are shown for $n = 8, 3$ in Fig. 3c, g, respectively.

To unravel the physical content of these encodings, we query the outputs of our algorithm. The code statistics in Fig. 3d reveal striking features: In class-8, of the sixteen 4-digit binary codes, only eight are ever assigned to $\mathcal{H}^8$, with half of the codes unused. Yet a 3-digit binary encoding, which has exactly eight available codes, is sub-optimal (Fig. 3b). Moreover, all eight codes have identical probability $1/8$ (Fig. 3d). This is in contrast to what happens in class-3, where the optimal compression uses three 2-digit binary codes, where only two codes are equiprobable, and the remaining one has probability $1/8$ (Fig. 3h). These puzzling results indicate that the optimal compression finds structure beyond merely the number of states of the DOF, which is essential to correlations with $\mathcal{E}$, and which is impossible to encode using fewer bits.

The used codes (that is, the $n$ states of effective DOF $\mathcal{H}^n$) are not arbitrary, but are related to the local symmetries of the super-quasilattice. This can be seen by investigating the structure of codes and the $\mathbf{\Lambda}^n$ maps. Notice, for example, that the eight 4-bit states of $\mathcal{H}^8$ can be arranged on a closed 8-cycle, such as (Fig. 3e)

$$1101 \rightarrow 1100 \rightarrow \cdots \rightarrow 1001 \rightarrow 1101, \qquad (4)$$

where each code has exactly two 1-bit-distant neighbours. Interestingly, this solves the four dimensional 'coil in the box' problem familiar from coding theory. Together with the equiprobability of these codes in Fig. 3d, this cyclic structure hints at a symmetry.

Indeed, a class-8 patch $V^8$ of the AB quasilattice is locally symmetric under $\pi/4$ rotations. The mutual information should be invariant under the action of this symmetry on the compression map $\mathbf{\Lambda}^8$.

Such rotations transform the linear filters $\Lambda_k^8$ via a permutation and inversion of the components, as can be verified visually in Fig. 3c:

$$C_8 : \left(\Lambda_1^8, \Lambda_2^8, \Lambda_3^8, \Lambda_4^8\right) \rightarrow \left(\Lambda_4^8, -\Lambda_3^8, -\Lambda_1^8, -\Lambda_2^8\right), \qquad (5)$$

which is a representation of a generator of the cyclic group $C_8$. We emphasise that it is now the compression map, and consequently the collective DOF now carrying a representation of what is a priori a (local) symmetry only of the AB tiling.

Strikingly, when we apply the transformation Eq. (5) to the states of the coarse grained DOF $\mathcal{H}^8$ (where it now amounts to the permutation of the binary digits and bit-flips), we find that it generates exactly the 8-cycle of Eq. (4). Since the 1-bit-flip transitions on this cycle are directly induced by $\pi/4$-rotations, the eight states of $\mathcal{H}^8$ can be aligned with the spatial orientations along the eight links of the 8-supervertex, as shown in Fig. 3i. This establishes $\mathcal{H}^8$ as a $\mathbb{Z}_8$ clock variable.

A similar analysis can be performed for other classes of $V^n$, which have a mirror symmetry. In particular, under the mirror reflection of the class-3 patch $V^3$, the two digits of $\mathcal{H}^3$ are swapped as (see Fig. 3g)

$$\text{Mirror} : \left(\Lambda_1^3, \Lambda_2^3\right) \rightarrow \left(\Lambda_2^3, \Lambda_1^3\right). \qquad (6)$$

Since the mirror axis is along the edge connecting the 8- and 3-vertices (see Fig. 3i), it associates the swap-invariant state $11$ with the edge pointing towards the 8-vertex, and the remaining equiprobable states $01$ and $10$ with the other two edges. Like in class-8, its transformation under symmetries establishes $\mathcal{H}^3$ as a 3-state clock variable, whose states can be identified with the super-quasilattice edges.

Hence, we see that the DOFs of the dimer system remain discrete under coarse graining. In particular, we compressed the dimer microstates on the microscopic links into $\mathbb{Z}_n$ clock variables that live on the *vertices* of the underlying super-quasilattice, where they mimic

the local symmetries. We found that this result holds equally at both $\sigma^{-2}$ and $\sigma^{-4}$ scale transformations, providing the first indication of a DSI. The persistent discreteness of the collective variables is to be contrasted with the situation on periodic lattices, such as the dimer coverings of the square lattice, which has an emergent continuous U(1) symmetry[35].

## Binding of clock variables into emergent super-dimers

Having identified the collective clock DOFs in different classes of blocks individually, we now turn to their correlations, where DSI is manifested fully. To this end, we simultaneously coarse grain dimer configurations in multiple blocks. Deflating the canonical blocks $V^n$ using trained compression maps (Fig. 3c,g), the correlations of the collective variables $\mathcal{H}^n$ indicate that the effective renormalised model of the clocks has hard-core attractive and repulsive interactions along the links of the AB super-quasilattice (see the bold edges in Fig. 2a).

We probe the correlations by conditioning on the state of one of the vertices. In Fig. 4b, c, fragments of $\sigma^{-2}$ and $\sigma^{-4}$ super-quasilattices are shown, with the state of the conditioning variable, identified with a direction, in orange and the conditional distribution of DOFs at the other vertices in grey. Remarkably, this distribution is very strongly correlated, effectively forcing occupation of some states and excluding others. To wit, when the 3-vertex DOF points towards the 8-vertex, the distribution $P(\mathcal{H}^8|\mathcal{H}^3)$ of the latter is sharply peaked in the matching direction, while no other neighbour of the 3-vertex points towards it (allowing, for example, the identification of the 8-vertex code `1011` with a specific spatial orientation in Fig. 3i). Conversely, when the 3-vertex DOF points towards one of its other neighbours, it is 'matched' by the latter, while the 8-vertex DOF distribution has zero weight precisely and only in the direction towards that 3-vertex.

Examining all such correlations, we arrive at a striking conclusion: the effective DOFs in $V$s throughout the quasilattice are paired with one and only one of their neighbours into emergent *super-dimers* on the edges of the super-quasilattice. The exclusion of certain clock variable orientations in Fig. 3a–e is a precise reflection of the hard dimer-constraints, which these super-dimers obey. Moreover, comparison of further correlations to those of the microscopic dimers in

Fig. 4a reveals that not just the local-dimer constraints, but also longer-range correlations are reproduced correctly. The physics of the microscopic dimer model on the AB quasilattice is thus replicated, to a high degree of accuracy, at the $\delta^{-2}$ scale and, again, at the $\delta^{-4}$ scale (where 'locking' is even sharper, see Fig. 4c), thereby demonstrating DSI across three scales.

## Discussion

Guided by the outputs of the RSMI-NE algorithm, we have seen how the quasiperiodicity of the AB quasilattice and the hard-core interactions of the dimer model conspire to *re*create self-similar DOFs at a higher scale, giving rise to DSI. A parallel work[28] gives a microscopic interpretation of the super-dimers as alternating dimer paths with respect to a certain reference configuration, and studies the criticality numerically.

Emergent continuous scale invariance is a standard signature of critical phenomena, being one aspect of conformal symmetry[46]. Here we instead encounter an exotic kind of critical phenomenon where this continuous conformal symmetry is broken to a discrete subgroup, thereby complicating the usual effective continuum theory description at large scales. This is particularly interesting, as it appears to challenge the received wisdom that quasicrystallinity should always be RG irrelevant[47].

We would like to also emphasise the dual computational and conceptual aspect of this result: In particular for the $\sigma^{-4}$ scale transformation, the RSMI-NE algorithm successfully encodes the symmetries and large-scale correlations in approximately $2^{10^3}$ dimer microstates into a highly structured linear coarse graining map $\Lambda^n$, which is effectively impossible to guess or analyse by human intuition only. Our demonstration of this approach on an open problem shows how machine learning tools, when paired with a physically motivated objective function, can bridge gaps between complex data and formal physical understanding.

In the Supplementary Information we have also applied the graph RSMI-NE algorithm on a class of non-bipartite random graphs, where we show that it can be used to construct a global order parameter for the frustrated Ising antiferromagnet. From the dual perspective of

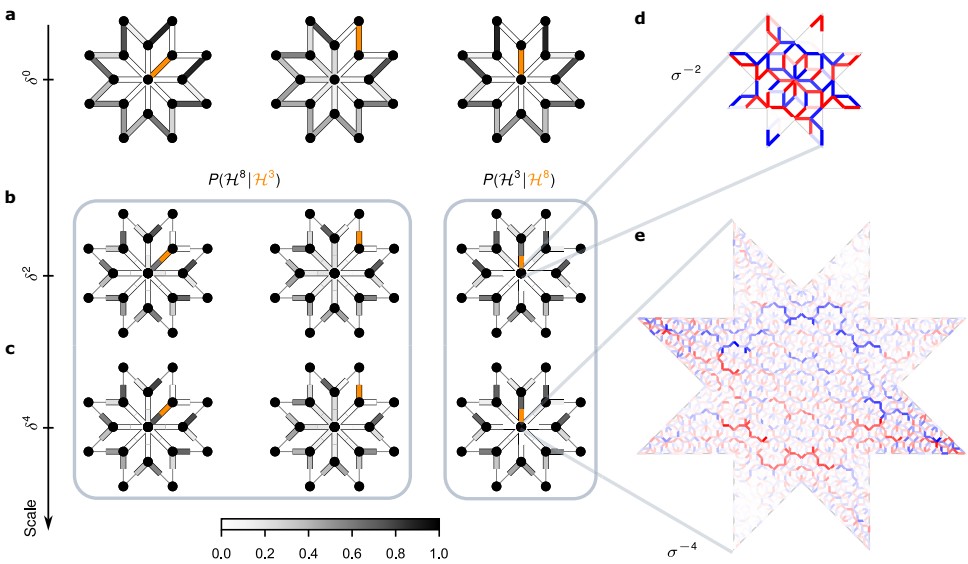

**Fig. 4 | Emergent dimer exclusion rule and self-similar dimer-dimer correlations across scales. a** The probability distribution of microscopic (i.e. $\delta^0$) dimers (in greyscale) on an AB patch, conditioned on one of the links (in orange) hosting a dimer. **b, c** First two columns: the probabilities $P(\mathcal{H}^8|\mathcal{H}^3)$ of the emergent clock variables on the $\delta^2$ and $\delta^4$ super-quasilattice (in greyscale), conditioned on two distinct states of one of the 3-clocks (in orange). The third column shows

distributions $P(\mathcal{H}^3|\mathcal{H}^8)$ conditioned on a state of the central 8-clock. Binding of adjacent clock variables into super-dimers obeying dimer exclusion constraints is revealed by sharply peaked conditional distributions. The effective super-dimers reproduce also longer-range dimer-dimer correlations at both $\delta^2$ and $\delta^4$ scales. **d, e** Examples of optimal coarse-graining filters producing the central 8-state clock variable at scales $\delta^2$ and $\delta^4$. The latter comprises 2760 microscopic links.

combinatorial optimisation, this amounts to solving the well-known optimal graph-bipartitioning problem in combinatorial optimisation and graph theory[48–51]. These results illustrate the practical applicability of our method—formally apparent from its mathematical definition—to more generic graph topologies, and its independence of the knowledge, or even existence, of preferred spatial blocks.

We therefore conclude that lossy compression allows effective DOFs to be extracted from the structure of information inherent in raw high-dimensional data, and that this approach excels in systems with non-regular geometry. Given that such cases are the norm rather than the exception in real-world applications, we expect compression theory to become an essential tool in the physical sciences.

## Methods

### Real-space mutual information based coarse graining

The method used to construct the effective DOFs is a generalisation of the compression theoretic approach first introduced by some of the authors in ref. 16 for translation invariant systems. The key insight is that the random variables, such as the local variable being coarse-grained $\mathcal{V}$ and its environment $\mathcal{E}$, in variational principle in Eq. (2) can be defined in any metric space, and need not be restricted to regular lattices as in ref. 16. Here we use the graph distance which provides a meaningful measure for spatial length scale on graphs with a local structure. This allows to define the optimal coarse-graining in any geometry, addressing one of the main challenges to applying RG in inhomogeneous systems.

The above construction is also useful in tackling a second key issue in RG approaches on disordered systems: One has to deal with the changing coarse-grained graph across different scales[4]. Here, the effective coarse-grained graph structure is defined by the correlations of the collective DOFs themselves. Thereby the procedure takes account of both the topology of the space and the interactions. Endowed with these two properties, the compression principle in Eq. (2) yields a mathematical definition of optimal coarse graining for inhomogeneous systems with local structure.

In concrete terms, our compression method is defined as follows. Consider a system of microscopic DOFs living on the graph $G$, defined, as usual, as the sets of vertices and edges. The physical space of configurations of the system living on the graph is described by a (large dimensional) random variable $\mathcal{X}$ distributed according to some joint probability distribution $P(\mathcal{X})$. The DOFs may exist either on the vertices, on the edges (as in the dimer model), or both.

Let further $G = \bigcup_i V^i$ denote a decomposition of $G$ into a set of simply connected local subgraphs (patches). A coarse graining of a partition $\mathcal{X} = \bigcup_i \mathcal{V}^i$ of the physical configurations into new variables $\mathcal{X}' = \bigcup_i \mathcal{H}^i$ is then defined as a conditional probability distribution

$$P(\mathcal{X}'|\mathcal{X}) = \prod_i P_{\Lambda^i}(\mathcal{H}^i|\mathcal{V}^i), \tag{7}$$

where each factor is an individual coarse graining of block variable $\mathcal{V}^i \mapsto \mathcal{H}^i$. This will be a compression map by construction, so it monotonically reduces the entropy $H(\mathcal{H}^i) \le H(\mathcal{V}^i)$. We describe a specific ansatz for such mappings below.

We emphasise the distinction between the spatial patch $V^i$, and configurations of DOFs supported on this patch $\mathcal{V}^i$, which is a random variable. The patch $V^i$ can be chosen as any local subgraph, for instance a topological ball defined using graph distance, or another set dictated by the structure of the problem, such as the tiles we used in the AB example (Fig. 2a).

Algorithmically, the static graph structure allows us to define a constant indexing of the individual DOFs that is fixed across all real-space samples. Therefore, once we use the graph structure to define the subsystems $\mathcal{V}^i, \mathcal{E}^i$ we then can forget about the connectivity of the subgraphs $V^i$, $E^i$ and simply cast the DOFs into vectors:

$$\underbrace{\left\{\mathcal{V}^i_j\right\}}_{\text{set of DOFs}} \xrightarrow[\text{permutation}]{\text{fixed}} \underbrace{\left[\mathcal{V}^i_j\right]}_{\text{vector of DOFs}}. \tag{8}$$

Therefore, we can use standard NN architectures to handle the physical configurations on the graph. Note that would not be able to use this trick in the more general case of dynamic graphs, as it requires respecting the permutation invariance of the vertices.

Maximisation of the real-space mutual information (RSMI)

$$I(\mathcal{H}^i : \mathcal{E}^i) = \mathbb{E}_{P(\mathcal{H}^i, \mathcal{E}^i)}\left[\log P(\mathcal{H}^i, \mathcal{E}^i) - \log P(\mathcal{H}^i)P(\mathcal{E}^i)\right]$$

between $\mathcal{H}^i$ and its distant environment $\mathcal{E}^i$ provides a variational principle for the coarse graining map $\Lambda^i$. The objective depends on the coarse graining mapping via the compressed $\mathcal{H}^i$ variables: $I(\mathcal{H}^i : \mathcal{E}^i) \equiv I_{\Lambda^i}(\mathcal{H}^i : \mathcal{E}^i)$. The construction of the RSMI objective function enables distiling the most relevant large-scale features[13,15], as it tracks the correlations with a distant environment $\mathcal{E}^i$. Formal connections between this objective and the most relevant operators in critical lattice systems has recently been demonstrated both numerically and analytically[13,14].

The compression is enforced by limiting the information capacity of $\mathcal{H}^i$ using a predetermined number of encoding bits (as we describe below), thereby directly implementing the rate constraint in the information bottleneck problem[12,13]. Note that the compression is informed both of the underlying graph structure, and of the physics of the model living on it, which are encoded in the statistics of the configuration samples $(\mathcal{V}^i, \mathcal{E}^i)$.

### Estimation of mutual information

The computationally challenging RSMI variational principle can be efficiently implemented with differentiable lower bounds on mutual information[17,18]. Such bounds are parametrised by a deep NN, which we call the *neural critic* function (see below). The key idea behind this approach is that the estimation of mutual information $I(\mathcal{X} : \mathcal{Y})$ is converted to a classification task; where the *neural critic*, $f(\mathcal{X}, \mathcal{Y})$, is trained to distinguish so-called *positive* and *negative* sample pairs, which are sampled respectively from the true joint distribution $P(\mathcal{X}, \mathcal{Y})$ and the product of marginals $P(\mathcal{X})P(\mathcal{Y})$[19].

Some of the authors have recently used these techniques to develop a tractable implementation of the variational principle in Eq. (2) on regular lattices[14,20]. This is the RSMI-NE algorithm, where the neural critic $f$ is optimised simultaneously with a coarse graining ansatz $\Lambda$ using stochastic gradient descent, as we describe below.

In the present work we extended this compression framework for RG, and the RSMI-NE package to systems on arbitrary static graphs by casting the configurations into vectors according to the fixed coordinate system defined by the graph. The graph-enabled RSMI-NE code using the NetworkX backend[52] is available publicly[21].

### The coarse-graining and neural critic ansatze

We specify the coarse graining $P_{\Lambda^i}(\mathcal{H}^i|\mathcal{V}^i)$ using an inner-product ansatz

$$\mathcal{H}^i_k := \tau\left(\Lambda^i_{kj}\mathcal{V}^i_j\right), \tag{9}$$

parametrised by a vector of linear NNs $\Lambda^i = (\Lambda^i_k)$, and $\tau$, which is a binary discretisation map (e.g. sign function). The index $k$ runs over the components of a coarse-grained variable, and the index $j$ refers to the spatial positions in region $\mathcal{V}$, where the indexing is defined with respect to a fixed labelling of vertices in the graph. Though we considered scalar DOFs in the AB dimer system, vector DOFs $\mathcal{V}^i_{jl}$ can also be handled by increasing the rank of the coarse graining filter by one: $\Lambda^i_{kjl}$.

In general, non-linear NN ansatze can also be used, and may be even necessary in certain cases, see for example ref. 53. Our code supports such more general mappings that do not have this multilinear structure, and are instead parametrised by deep NNs.

We implemented the binary mapping τ using the Gumbel-softmax trick[54], which is an annealed relaxation of the Bernoulli distribution for $\mathcal{H}_k^i$. This allows us to backpropagate through the discrete sampling process, and to train the coarse-graining filters using stochastic gradient descent, while ensuring a fixed rate of compression via discretisation. We used an annealing schedule exponential relaxation rate of $5 \times 10^{-3}$, so that the variables are effectively binary at the end of the training.

The critic function in the variational RSMI lower-bound is implemented using a separable architecture

$$f(\mathcal{H}, \mathcal{E}) = u(\mathcal{H})^{\mathrm{T}} v(\mathcal{E}) \tag{10}$$

where we used two-layer deep NNs for $u$ and $v$, with hidden dimension 16 and output dimension 8 (the hidden dimension is contracted in the product of the two networks).

### Training details

We trained the NNs using stochastic gradient descent with learning rate of $10^{-3}$ using 50,000 sample dimer configurations, generated via the directed-loop Monte Carlo algorithm on the AB graph. The total graph we considered contains 26,177 nodes (the full graph is shown in the Supplementary Information). The sample dataset is supplied to the RSMI-NE algorithm in mini-batches of size 1000 and 120 epochs of the entire dataset.

The coarse-grained block variable $\mathcal{V}$ at a given scale $\delta^s$ is defined on the $\sigma^s$ inflated tiles $V$ shown with different colours in Fig. 2a. The corresponding environment regions $E$, are defined as a shell with radius given by a fixed graph-distance from the centre of $V$. In particular for $\delta^2$, $E$ is defined by an inner radius $L_{E_{\mathrm{in}}} = 9$ and outer radius $L_{E_{\mathrm{out}}} = 24$, whereas for $\delta^4$ we used $L_{E_{\mathrm{in}}} = 40, L_{E_{\mathrm{out}}} = 64$, as shown in Supplementary Fig. 2. Examples of the corresponding $\sigma^{-4}$ coarse-graining filters are shown in Supplementary Fig. 3.

## Data availability

The data generated during the course of this study have been deposited in the *Figshare* repository at https://doi.org/10.6084/m9.figshare.27245481(ref. 55).

## Code availability

The RSMI-NE software used in this study is available as an open-source repository in the *Zenodo* repository linked in ref. 21 and https://github.com/RSMI-NE/RSMI-NE.

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

## Acknowledgements

D.E.G. gratefully acknowledges support from the Simons Foundation and NSF through the National Institute for Theory and Mathematics in Biology. D.E.G. and S.D.H. acknowledge financial support from the Swiss National Science Foundation Grant No. 182240. S.B. acknowledges support by the European Research Council under the European Union Horizon 2020 Research and Innovation Programme via Grant Agreement No. 804213-TMCS. Z.R. acknowledges support from ISF grant 2250/19. F.F. acknowledges support from EPSRC Grant No. EP/X012239/1. M.K.-J. gratefully acknowledges financial support from the European Union's Horizon 2020 programme under Marie Sklodowska-Curie Grant Agreement No. 896004 (COMPLEX ML).

## Author contributions

D.E.G., S.B., S.D.H., Z.R., F.F. and M.K.-J. designed the research, performed the research and wrote the manuscript.

## Competing interests

The authors declare no competing interests.
