## [Transparent Peer Review file · Nature Communications]

Compression theory for inhomogeneous systems

Corresponding Author: Dr Felix Flicker

Version 0:

Reviewer comments:

Reviewer #1

(Remarks to the Author)

This manuscript extends (some of) the authors' previous important work on information-theoretic formulations of the renormalization group to spatially inhomogeneous systems. This extension is applied to dimer models on quasicrystals, and yields physically interesting results (though I think it is an overstatement to claim -- as the authors do -- to "prove" a conjecture, unless their results are more mathematically rigorous than I understand).

One general question: while the information-theoretic machinery is very nice, it is not clear to me how much of the work is being done by the precise choice of spatial blocks. Ideally this technique should work even if one doesn't know the underlying structure of the problem (in this case the blocks).

I think the work is clearly novel and interesting enough to be published.

Reviewer #2

(Remarks to the Author)

As described in the abstract of this manuscript, the goal of the authors is to develop a compression method for complex systems and "extract relevant degrees of freedom in arbitrary geometries." The main part of this work focuses on applying an existing algorithm "RSMI-NE" (acronym to be defined in the manuscript) to one specific graph, the Ammann—Beenker (AB) model. The authors use their algorithm to extract two known symmetries (rotation and reflection) associated with two elementary patches and they provide insights into the discrete scale invariance of the considered system.

Based on the findings of the present analysis, the manuscript lacks the necessary evidence to substantiate its claims—specifically, its ability and utility in extracting relevant degrees of freedom across a wider range of graph structures commonly studied in the field of complex systems. Given the mismatch between the overall goal of this work and the focus on a single graph structure and model, I do not find the presented results significant enough to justify publication in Nature Communications.

Additional points:

(1) page 1: The beginning of the section "The system" mentions the AB model, but it doesn't explain why the authors decided to focus on this specific model. The introduction is written in relatively broad terms discussing general complex systems, biological tissue, and graph representations. Then there seems to be a sudden switch to the AB model. The rationale behind studying this particular system should be described in more detail.

(2) page 2: Please provide a clear definition of discrete scale invariance, e.g., in the fourth paragraph.

(3) page 3: The use of the term "class" in different (?) contexts makes some parts of the text difficult to follow. For example, the authors describe that Fig. 2 shows four "classes" of elementary blocks. The next sentence then begins with the term "inequivalent class", raising the question if this refers to one of the four classes or something else. The same sentence ends with defining κ_n as the connectivity, or class, of \mathcal{V}_n .

The example in the following sentences covers classes 8 and 3, which are shown in Fig. 3(c,g). But how are these classes connected to the prior paragraphs mentioned in my above comments. Maybe add "green" and "blue" if the corresponding

"patches" in Fig. 2 are meant?

(4) The terms "exotic critical point" and "conformal symmetry" appear in the abstract, but are not picked up again later on in the manuscript. This part in the abstract should be better connected to the last part in the results section. I think that it would be also beneficial for the manuscript to briefly outline its structure at the end of the introduction. Right now, some parts of this work are not very well connected.

(5) In my opinion, the description of the compression algorithm should be part of the main text as it is central to the whole manuscript. In the current manuscript, one has to continuously switch between the main text and Appendix A. For example, there has been no mention of "neural network" function approximators until page 3 (results section). This part becomes clearer in Appendix A. I would recommend clarifying the whole algorithm in the main text and leaving the paragraphs on specific parameter choices in the appendix. An algorithm flowchart may be helpful too.

Minor issues:

- * Please define the acronym "RSMI-NE" at its first occurrence
- * "a RG analysis" → "an RG analysis"
- * "strongly correlated" instead of "strongly interacting" in the abstract? The later is used on page 2.
- * "outputs of our algorithm A parallel work" → "outputs of our algorithm. A parallel work"

Reviewer #3

(Remarks to the Author)

The paper under review concerns itself with a previous proposal (put forward by several of the present authors; references [15--17]) to identify the effective degrees of freedom in real-space (spin-block) renormalization using information-theoretic criteria: the coarse-graining transformation in each block is chosen to maximize a particular mutual information. It was previously shown that, when applied to standard examples, such techniques produce effective degrees of freedom with short-range interactions. In general, the proposal fits into the current activity surrounding the relationship between geometry and entanglement, correlation, or other information-theoretic measures that is visible in many different areas of theoretical physics, from high-energy theory on up.

In the paper at hand, numerical techniques (a neural network) are applied to explore the proposed definition in the context of a particular dimer model, defined on a class of aperiodic tilings (the Ammann-Beenker tilings) with discrete scale invariance. Numerical results indicate that the degrees of freedom after one coarse-graining transformation are again described by precisely the same class of dimer configurations. (More precisely, the degrees of freedom on each new vertex correspond to a choice of half-edge, and their correlations localize to dimer configurations in which adjacent half-edges join together into dimers.)

These numerical results suggest the existence of an exact self-similarity transformation that acts on dimer models and produces a fixed point of the real-space renormalization group. This is indeed the case, and the corresponding transformation was constructed in a companion paper (Biswas and Parameswaran, reference [40]).

The paper is interesting, its numerical findings are worth reporting, and it is an important piece of a larger story the authors are telling. However, I cannot help but feel that the way in which the paper is written detracts from the results in at least two significant ways: Firstly, the conceptual value of the numerical techniques being used---which is nontrivial!---is dramatically oversold. Secondly, and relatedly, insufficient care is paid to a clear and transparent explanation of precisely how the example under study differs from standard examples and ideas in the literature. As a result, it is easy for the reader to miss one of the most interesting features of the example: the lack of any (obviously visible) emergent symmetry at the RG fixed point.

Perhaps the most important point is the following: Much of the motivational discussion has to do with "generic" or "irregular" geometries, but the models considered are, in a crucial way, *not* generic. They admit an obvious action of discrete scale transformations, given by the "inflation" and "deflation" transformations, which give a unique way to block an AB tiling into a larger-scale tiling of the same kind.

Similarly, it is claimed that the discrete RG transformation is defined "in a manner that is independent of geometry." Locally, this is true: the procedure that takes place *within* a fixed spin block is defined purely in terms of mutual information. But there are *many* possible ways of blocking spins on a generic graph, and there is certainly no canonical choice. Is there any sense in which the outputs of the procedure (which are then defined on *different* replacement graphs) are insensitive to the choices made? For the AB tilings, there is a canonical choice, because the group of discrete scale transformations acts naturally by "deflation" on the set of such geometries. But this means that geometry *is* playing an essential role in the definition of the procedure.

I have a similar gripe about the way that inhomogeneity is discussed. This is the important new feature of these tilings. But the authors should make it more clear that the translation invariance or geometric systems of field theories, or spin systems on periodic graphs, serves a separate purpose to the action of scale transformations. The latter are essential to define the renormalization group; they are present in the authors' example, and are used in exactly the same way. The former allow one to separate modes by wavelength, and to define low-energy effective degrees of freedom by restricting attention to long-wavelength modes. (Wavelength and energy are normally connected by some sort of dispersion relation.)

The mutual-information procedure gives a clean, conceptually nice, and broadly applicable way of thinking about identifying good low-energy degrees of freedom. It allows one to do this without making reference to wavelength. But, since scale transformations are still present, one could certainly ask about the spectrum of scaling dimensions---about energies directly, rather than via the proxy of wavelength. What relationship would such a prescription stand in to the mutual-information procedure? This is, of course, not the topic of the present paper, but it deserves comment; the procedure being studied is of interest, and can be justified carefully and clearly, so the authors should do so.

One could also point to the abstract, where it is claimed that "...the vastness of configurational spaces present[s] a fundamental challenge to theoretical tools, such as the renormalization group, which were so successful in characterizing the universal physical behaviour in critical phenomena." This vastness is common to *every* example where the renormalization group is studied---every quantum field theory, and every statistical mechanics model. So, to the extent that these tools were successful, it is clearly not the fundamental challenge.

Lastly, at the end, the authors claim that we "have, in effect, reached a point where ML techniques can not only assist, but facilitate progress in theoretical physics." I fail to see how anything in the present paper has anything to do with this claim---which should not be understood as a criticism of the work done in the paper, but rather as evidence that the claim is out of place in this article. The authors explore a proposed construction of effective degrees of freedom which is computationally unwieldy; they use a computer and a specific kind of software to do this; they produce numerical output which is interesting, and leads to a conjecture, which they then go on to prove (in the companion work [40]). How is this, operationally or qualitatively, any different than the fact that the conjecture of Birch and Swinnerton-Dyer was formulated based on numerical evidence that was produced with the help of a computer? That happened in the early 1960s. The paper's relationship to [40] should also be articulated more clearly earlier on in the text.

In conclusion, I can only recommend publication of this paper after serious revision, to clarify precisely what is done in it, chart its relationship to the authors' other related publications, to better highlight the results and their physical and conceptual importance, and to eliminate unnecessarily florid, misleading, or unclear language wherever possible.

Version 1:

Reviewer comments:

Reviewer #1

(Remarks to the Author)

The authors have addressed the main issue that I (and other referees) raised in the previous round -- namely that the model seemed fine-tuned in a way that made it easy to guess the right block structure. The new version contains appendices that extend the results to other models with less exploitable structure. I think this addresses the main criticism and I recommend publication.

(Remarks on code availability)

Reviewer #2

(Remarks to the Author)

I thank the authors for their response. Overall, an existing method (cf. Refs. 17–19 and 29) has been applied to a known problem with a solution that has been derived in a separate work (cf. Ref. 26). While the authors have included an additional example in the Appendix, I find the overall contribution unconvincing. Furthermore, many parts of the manuscript remain unclear. This lack of clarity is primarily due to insufficient communication of the actual contributions and presentational issues regarding the employed algorithm.

(1) The contributions of this paper remain unclear. It primarily applies an existing method (cf. Refs. 17–19), but this is not clear from the revised introduction. In the first two pages, the focus is mainly on an algorithm overview, which suggests that this is a main contribution of the paper.

Additionally, the authors' explanation in their response letter that the employed numerical methods, particularly the machine-learning approaches, are subordinate is confusing. If these methods are not central, why do the authors state in the introduction that advancements in machine learning are a key part of their method without further detailed explanation? It would be much clearer to explain at the very beginning that this paper focuses on applying an existing method to problems whose solutions have already been discussed in prior work.

(2) The revised method description on pages 1 and 2 is still difficult to follow. The method that the authors employ is a key part of the paper, but the reader is not properly introduced to it. One has to continuously switch between this part, other parts of the manuscript, and Refs. 17–19. For example, the authors mention subgraphs V^i without defining the actual graph. Mutual information appears in the new Eq. (2), but it is neither properly defined nor adequately explained. Instead the authors refer to their prior papers on this topic (Refs. 17–19 and 29). On page 2 the authors mention that they will not further distinguish between "the local graph" and "the configurations" because the "correct meaning is implied by the context". The term "local graph" appears once in the whole manuscript. Is it the same as V^i ? Overall, this revised section is not clear.

(3) The main objective of the new appendix section is not well-motivated. The authors apply the previously introduced method to the Ising model on an almost bipartite graph. As described by the authors, the problem of finding the ground state is tractable for a bipartite graph. Keeping the graph close to a bipartite one (as in the added simulation results) ensures that the problem is simple enough. The authors then demonstrate that their method can, e.g., correctly identify the partitioning and solve the minimum bisection problem, given an almost bipartite graph.

(a) A comparison with existing (exact and approximate) graph optimization approaches is missing. Based on the current results, it isn't clear if the presented method holds any computational advantage over existing graph-bisection approaches. What is the size of the considered graph and what is the computational complexity of the employed algorithm?

(b) What would the order parameter dependence on β look like if one (i) wouldn't employ any coarse graining or (ii) combine an existing bisection approach with some kind of naive coarse-graining baseline? Such comparisons can help provide a mechanistic interpretation of $\mathcal{O}(\lambda)$ in the considered example.

(4) The vertical error bars in Fig. S5 are not defined.

Minor:

Reference 21: "Nonequilibrium fixed points of coupled Ising models" → "Nonequilibrium fixed points of coupled Ising models"

Reference to the following paper is missing: Wolff, U. (1989). Collective Monte Carlo updating for spin systems. Physical Review Letters, 62(4), 361.

(Remarks on code availability)

Reviewer #3

(Remarks to the Author)

I thank the authors for their careful attention to the various referee reports, and to the thought put into the revisions. I find the new version of the article substantially improved in exposition. It furthermore represents an improvement in content, since the new Appendices B and D provide explicit and illustrative (if concise) information about applications of the algorithm to graphs that are not acted on by a group of discrete scale transformations.

Other than the new appendices, perhaps the most substantial changes are to the introductory portion of the article; the first two pages have been largely rewritten. In my opinion, the changes make the paper clearer to read and outline, much more transparently, what its central results are and how these are to be contextualized--both amongst other results and as related to large-scale motivating questions.

I have no further concrete suggestions for the improvement of the paper, and am willing to recommend its publication in Nature Communications if the other referees concur.

(Remarks on code availability)

We thank the Referee for the endorsement and for their very detailed and insightful comments.

We address the Referee's present detailed points below:

Referee #1: This extension is applied to dimer models on quasicrystals, and yields physically interesting results (though I think it is an overstatement to claim – as the authors do – to “prove” a conjecture, unless their results are more mathematically rigorous than I understand).

Authors: We agree with the Referee that the term “prove” in a strict mathematical sense might be too strong, as our method, while constructive and interpretable is nevertheless numerical. We have changed the phrase “constructively proving an open conjecture...” into “explicitly constructing emergent DOFs that confirm an open conjecture...” in the revised version.

Referee #1: One general question: while the information-theoretic machinery is very nice, it is not clear to me how much of the work is being done by the precise choice of spatial blocks. Ideally this technique should work even if one doesn't know the underlying structure of the problem (in this case the blocks).

Authors: We thank the Referee for this comment. This is an important point, and, in fact, showcases the generality of our method, as the results are robust to the details of the spatial block choice. In the revised manuscript we showcase this generality and robustness in two ways: first, we have added the new Appendix D, where we show that the RSMI-NE algorithm can reproduce the conclusions regarding the discrete-scale invariance of the dimer model on AB tiling, without a fine-tuned choice of the blocks that are informed by the underlying structure. To this end we choose the spatial blocks as topological balls of fixed radius and show the same spatial structures of the solutions emerge. Furthermore, in reply to the other Referees' comments we have applied the RSMI-NE to a frustrated antiferromagnet on random graphs devoid of any quasiperiodicity, where a canonical choice of block shapes does not even exist.

Referee #1: I think the work is clearly novel and interesting enough to be published.

Authors: We thank the Referee for the endorsement. We are glad that the Referee found the work novel and interesting enough to be published in *Nature Communications*.

We thank the Referee for the report. We agree that certain points of the presentation could be improved, in particular by addition of new examples demonstrating explicitly the applicability of our method beyond the context of quasiperiodic systems. We address the Referee’s detailed points below:

Referee #2: As described in the abstract of this manuscript, the goal of the authors is to develop a compression method for complex systems and “extract relevant degrees of freedom in arbitrary geometries.” The main part of this work focuses on applying an existing algorithm “RSMI-NE” (acronym to be defined in the manuscript) to one specific graph, the Ammann—Beenker (AB) model. The authors use their algorithm to extract two known symmetries (rotation and reflection) associated with two elementary patches and they provide insights into the discrete scale invariance of the considered system.

Authors: We feel that the above summary does not entirely correctly reflect our contribution, and we apologise that our writing failed to clearly convey that the generality of our method is a consequence of its definition, which does not depend on the details of the graph (see below). We clarified this by rewriting the introduction of our manuscript to include more details of our lossy-compression principle and the RSMI-NE algorithm, thereby also addressing the point (5) in Referee’s report.

Nevertheless, we agree with the Referee that the manuscript would benefit from the inclusion of another example. To this end, in the revised manuscript we have applied the algorithm to an entirely different system of frustrated antiferromagnets on random graphs devoid of any quasiperiodicity. These results are contained in Appendix B, further demonstrating the generality of our approach.

We would like to clarify that the methodological advance of our work is the realization that the compression-theoretic framework of coarse-graining used in the RSMI-NE algorithm, which we introduced earlier for systems on translation-invariant lattices, provides an avenue to **define** real-space RG on arbitrary graphs with local structure. No such general definition existed before and consequently no generally applicable numerical method existed either, as opposed to narrow solutions applicable in specific cases (e.g. “bond moving” [PRB **28**, 5022 (1982)]).

We have solved both of the above problems simultaneously, which is both an important theoretical and numerical advance. To showcase both the correctness of the theoretical idea as well as the efficiency of our numerical algorithm we applied it directly to an open problem, about which conjectures had recently been posed in the literature, specifically about the emergence of discrete scale invariance. This is an example of “exotic” critical point, as typically *discrete scale invariance of the lattice does not* imply discrete scale invariance of the physics of a model defined on this lattice; instead in the long-wavelength theory this is usually enlarged to a continuous symmetry, as is the case with the very same interacting dimer model on the equally discrete scale invariant standard square lattice.

Using the method we introduced we constructively show this theoretical conjecture to hold true. This is achieved by explicitly showing that the relevant collective degrees of freedom constructed by the

algorithm are “super-dimers”, reproducing correlations at lower scales, and thus manifesting discrete scale invariance of the physics of this strongly interacting system. We would like to clarify that we do not extract “two known symmetries associated with the local patches” – this would be entirely trivial. Instead, one of our results is that the emergent collective degrees of freedom supported in these patches carry the representations of these discrete local symmetries, which does not generically occur.

We also emphasize that the known symmetries of the *AB tiling* are insufficient to infer the nature of the collective degrees of freedom *of the model living on this tiling*, or their interactions, which is precisely why this was an open problem. This is a fundamental physical distinction, and we apologise if it was not sufficiently clear in the manuscript. The actual key finding of this study is that the correlations between the newly constructed collective DOFs mimic those of the microscopic dimers – this is the actual Discrete Scale Invariance (DSI) of the physics.

Referee #2: Based on the findings of the present analysis, the manuscript lacks the necessary evidence to substantiate its claims—specifically, its ability and utility in extracting relevant degrees of freedom across a wider range of graph structures commonly studied in the field of complex systems. Given the mismatch between the overall goal of this work and the focus on a single graph structure and model, I do not find the presented results significant enough to justify publication in Nature Communications.

Authors:

We are disappointed with the Referee’s conclusion, which we believe is primarily due to the shortcomings in our exposition, particularly our failure to clearly explain the RSMI-NE method in the manuscript.

As outlined in our response to the previous point, the RSMI-NE method is broadly applicable and not limited to the specific problem we applied it to. The chosen system is a strongly interacting model on a complex graph, a characteristic it shares with many challenging problems in complex systems and biology. Importantly, the system selected was not a trivial example but rather the subject of an open theoretical conjecture, which we successfully verified constructively. We firmly believe that application to an open problem is the ultimate test of any novel method.

Furthermore, in the revised manuscript we included results on a different class of systems on random graphs, devoid of any quasiperiodicity, which explicitly shows the method is general and applicable to generic graphs. We emphasize that this follows directly from the definition of the method, which works on any graph, and this general definition is also one of the key contributions of the paper, a point which was perhaps not sufficiently highlighted in the original manuscript. It is also important to note the contribution of our work in the form of an open source code (available in <https://github.com/RSMI-NE/RSMI-NE>) implementing our approach on generic graph geometries, which is behind the numerical results in this work (which we subsequently included in a new version of RSMI-NE package).

We hope that the Referee will find that the manuscript revisions improved clarity and addressed their concerns.

Referee #2: (1) page 1: The beginning of the section “The system” mentions the AB model, but it

doesn't explain why the authors decided to focus on this specific model. The introduction is written in relatively broad terms discussing general complex systems, biological tissue, and graph representations. Then their seems to be a sudden switch to the AB model. The rationale behind studying this particular system should be described in more detail.

Authors: We acknowledge the Referee's concerns about the clarity of our introduction.

The strongly correlated dimer system on the Amman-Beenker (AB) tiling is precisely an example of a system where interactions interplay with an inhomogeneous and complex graph structure, a feature it shares with the biological and other complex systems mentioned in the introduction. Crucially, however, this system was the subject of a precise and open theoretical conjecture about the nature of its effective degrees of freedom (the emergence of discrete scale invariance), which provides a perfect test for our method.

In the revised manuscript, we have enhanced the introduction by clarifying this point, as indicated in the highlighted sections.

Referee #2: (2) page 2: *Please provide a clear definition of discrete scale invariance, e.g., in the fourth paragraph.*

Authors: We have modified the corresponding paragraph by clarifying what is meant by discrete scale invariance.

Referee #2: (3) page 3: *The use of the term "class" in different (?) contexts makes some parts of the text difficult to follow. For example, the authors describe that Fig. 2 shows four "classes" of elementary blocks. The next sentence then begins with the term "inequivalent class", raising the question if this refers to one of the four classes or something else. The same sentence ends with defining n as the the connectivity, or class, of \mathcal{V} .*

The example in the following sentences covers classes 8 and 3, which are shown in Fig. 3(c,g). But how are these classes connected to the prior paragraphs mentioned in my above comments. Maybe add "green" and "blue" if the corresponding "patches" in Fig. 2 are meant?

Authors: We thank the Referee for pointing this out. We have clarified that the classes of coarse-grained patches are labelled by the connectivity of the supervertex where they live on. The use of the term "class" is then consistent across the text. In order to avoid the confusion that \mathcal{V} are not the classes of coarse grained blocks, but are random variables containing the degrees of freedom on such a block, we use the notation V to denote the *subgraphs* in the revised manuscript. Also, on page 4, in the paragraph starting with "Having found the emergent DOFs..." we have made the following change: "emergent DOFs in each class \mathcal{V} " \rightarrow "emergent DOFs in different classes of blocks \mathcal{V} ".

Referee #2: (4) *The terms "exotic critical point" and "conformal symmetry" appear in the abstract, but are not picked up again later on in the manuscript. This part in the abstract should be better connected to the last part in the results section.*

Authors: In the new penultimate paragraph, we have added a short discussion to connect our result about discrete scale invariance to the criticality of the system and broken conformal symmetry, and to

emphasise the physical importance of these features.

Referee #2: *I think that it would be also beneficial for the manuscript to briefly outline its structure at the end of the introduction. Right now, some parts of this work are not very well connected.*

Authors: We have added a brief outline of the paper just before the “Results” section.

Referee #2: *(5) In my opinion, the description of the compression algorithm should be part of the main text as it is central to the whole manuscript. In the current manuscript, one has to continuously switch between the main text and Appendix A. For example, there has been no mention of “neural network” function approximators until page 3 (results section). This part becomes clearer in Appendix A. I would recommend clarifying the whole algorithm in the main text and leaving the paragraphs on specific parameter choices in the appendix. An algorithm flowchart may be helpful too.*

Authors: We thank the Referee for raising this point. To address it, we have added more details about the RSMI principle and lossy compression in the main text, which we believe will make the general applicability of the RSMI-NE algorithm more evident for the reader. In the revised manuscript we have also added more details on how we use neural networks both for the estimation of mutual information, and as a coarse-graining ansatz for readers less familiar with the subject. Moreover, we have added more precise references to the Methods section (previously Appendix A) where the technicalities of the method are described.

We would like to clarify that the whole physical content of the work could, in fact, be understood solely via the solutions to the equation defining the compression problem, without any reference to how exactly numerically they are obtained, i.e. the Methods section. While of key importance for practical applications, the details of the numerical toolbox which we extended are nevertheless conceptually subordinate. Particularly, the specific neural network component behind information estimation and coarse-graining involved in practical implementation of the algorithm, which is also described in the Methods section (previously Appendix A), is shared with the regular lattice approach and has already been explained in detail in our previous published works [PRL **172**, 240603 (2021), PRE **104**, 064105 (2021)], including flow-chart style figures the Referee requests.

Referee #2: *Minor issues:*

- *Please define the acronym “RSMI-NE” at its first occurrence*
- *“a RG analysis” → “an RG analysis”*
- *“strongly correlated” instead of “strongly interacting” in the abstract? The later is used on page 2.*
- *“outputs of our algorithm A parallel work” → “outputs of our algorithm. A parallel work”*

Authors: We thank the Referee for pointing these text issues. We addressed all of these points in the revised version.

We thank the Referee for their very detailed and insightful comments which helped us improve the manuscript. We are very grateful for the detailed reading, and for the suggestions and interesting questions, all of which we address in detail below, and in the revised manuscript.

We address the Referee's detailed points below:

***Referee #3:** The paper under review concerns itself with a previous proposal (put forward by several of the present authors; references [15–17]) to identify the effective degrees of freedom in real-space (spin-block) renormalization using information-theoretic criteria: the coarse-graining transformation in each block is chosen to maximize a particular mutual information. It was previously shown that, when applied to standard examples, such techniques produce effective degrees of freedom with short-range interactions. In general, the proposal fits into the current activity surrounding the relationship between geometry and entanglement, correlation, or other information-theoretic measures that is visible in many different areas of theoretical physics, from high-energy theory on up.*

Authors: We thank the Referee for this insightful summary of our work and its context. We agree that the relationship between geometry and entanglement, correlation, or other information-theoretic measures is a very promising and exciting area of research, and we are glad that the Referee sees our work as a contribution to this field.

***Referee #3:** The paper is interesting, its numerical findings are worth reporting, and it is an important piece of a larger story the authors are telling. However, I cannot help but feel that the way in which the paper is written detracts from the results in at least two significant ways: Firstly, the conceptual value of the numerical techniques being used—which is nontrivial!—is dramatically oversold. Secondly, and relatedly, insufficient care is paid to a clear and transparent explanation of precisely how the example under study differs from standard examples and ideas in the literature. As a result, it is easy for the reader to miss one of the most interesting features of the example: the lack of any (obviously visible) emergent symmetry at the RG fixed point.*

Authors: We thank the Referee for his appraisal, recognizing both the importance and novelty of our detailed results, as well as their part in a burgeoning larger important research direction.

The Referee finds that some aspects of presentation detracts from the importance our results. We accept this criticism, and have implemented corresponding changes in the manuscript. Specifically, the Referee finds that the importance of the numerical aspect of our method, while nontrivial, is nevertheless oversold. We agree with the Referee that we might have erred on the side of enthusiasm in our exposition of this facet, and we have toned down certain statements in the introduction. We would like to try to convince the Referee, however, that the numerical methods which are currently at our disposal allow to directly attack large scale problems starting from essentially raw and high-dimensional experimental data (such as movies of cells or fluid flow), which was never possible before. While the

Referee is entirely correct in thinking that it is in some sense a quantitative change with respect to how numerical methods have been used in previous decades, the scale of that quantitative change makes a qualitative leap in capabilities possible. We discuss this in more detail in reply to one of the Referee's further points.

We also thank the Referee for pointing out the second aspect of the writing which insufficiently highlighted the nontrivial nature of the system we studied, and thus the importance of our results. The strongly correlated dimer system on the AB tiling is indeed highly unusual in that the discrete symmetries of local patches of the lattice and the discrete scale invariance of the lattice remain fixed also in the effective theory of the interacting system at the RG fixed point, and no enlarged emergent symmetry is generated. This is in contrast to the usual situation when the RG fixed point of a model defined microscopically on the lattice possesses continuous symmetry (as, for instance, even in the example of the very same interacting dimer model on the square lattice, where emergent U(1) symmetry appears). In the revised manuscript we have emphasized this point more strongly in the introduction by adding a new paragraph to discuss the significance of the lack of conformal symmetry for the present example.

*Referee #3: Perhaps the most important point is the following: Much of the motivational discussion has to do with “generic” or “irregular” geometries, but the models considered are, in a crucial way, *not* generic. They admit an obvious action of discrete scale transformations, given by the “inflation” and “deflation” transformations, which give a unique way to block an AB tiling into a larger-scale tiling of the same kind.*

*Similarly, it is claimed that the discrete RG transformation is defined “in a manner that is independent of geometry.” Locally, this is true: the procedure that takes place *within* a fixed spin block is defined purely in terms of mutual information. But there are *many* possible ways of blocking spins on a generic graph, and there is certainly no canonical choice. Is there any sense in which the outputs of the procedure (which are then defined on *different* replacement graphs) are insensitive to the choices made? For the AB tilings, there is a canonical choice, because the group of discrete scale transformations acts naturally by “deflation” on the set of such geometries. But this means that geometry *is* playing an essential role in the definition of the procedure.*

Authors: We thank the Referee for this insightful comment, which touches upon an important point highlighting, in fact, the generality and robustness of our approach.

As the question of generality of our algorithm is of key importance to our whole approach, we have furthermore decided to demonstrate it on systems on a different class of random graphs which do not possess any local symmetries or scale invariance. Specifically, in the new Appendix D we have applied the RSMI-NE to frustrated antiferromagnets on non-bipartite random graphs, where there is no canonical choice of coarse-graining blocks.

The Referee is entirely correct in pointing out that the AB tiling is indeed not generic, in that the discrete scale transformations are crucial for the definition of the quasiperiodic graph, which then brings about a canonical choice for the blocks. Crucially, however, the results of the RSMI-NE algorithm *do not* essentially rely on this choice. In the revised manuscript we added the new appendix D, where we

show that our algorithm can reproduce the conclusions regarding the discrete-scale invariance of the dimer model on AB tiling, by using blocks defined by topological balls of radius L_V around the sites of the AB tiling, without a fine-tuned choice of tiles informed by the underlying structure. We note that the local structure of the course-graining transformations’ optimal parameters is strikingly similar when using the canonical tiles or the topological balls (where they overlap). It is worth pointing out however, that using the canonical tiles is a form of supplying the algorithm with “prior knowledge”, which does improve the quantitative numerical performance, in that the correlations are reproduced more accurately. This is very much in spirit of the current trend of “physics informed machine learning (PIML)”, where generically incorporating the physical knowledge in the cost function, architecture, or data preparation improves performance of numerical methods.

It is also worth pointing out that the question can be turned around: The geometry of the blocks indeed does play an important role in the definition of the coarse-graining procedure. However, the success of the block structure can be quantified by the real-space mutual information. In other words, the value of maximal RSMI can be used as a diagnostic to determine the quality of the block structure proposed.

Referee #3: I have a similar gripe about the way that inhomogeneity is discussed. This is the important new feature of these tilings. But the authors should make it more clear that the translation invariance or geometric systems of field theories, or spin systems on periodic graphs, serves a separate purpose to the action of scale transformations. The latter are essential to define the renormalization group; they are present in the authors’ example, and are used in exactly the same way. The former allow one to separate modes by wavelength, and to define low-energy effective degrees of freedom by restricting attention to long-wavelength modes. (Wavelength and energy are normally connected by some sort of dispersion relation.)

The mutual-information procedure gives a clean, conceptually nice, and broadly applicable way of thinking about identifying good low-energy degrees of freedom. It allows one to do this without making reference to wavelength. But, since scale transformations are still present, one could certainly ask about the spectrum of scaling dimensions—about energies directly, rather than via the proxy of wavelength. What relationship would such a prescription stand in to the mutual-information procedure? This is, of course, not the topic of the present paper, but it deserves comment; the procedure being studied is of interest, and can be justified carefully and clearly, so the authors should do so.

Authors: We thank the Referee for this comment. Real-space mutual information indeed enables accessing the relevant large-scale degrees of freedom without making reference to proxies like wavelength. This is crucial for inhomogeneous systems, where the k-space RG is not useful, as organising the modes according to wavelength does not help targeting the low-energy degrees of freedom. On the other hand, the language of “low-energy modes” can also be misleading in certain contexts, particularly where energy may not be a directly applicable concept, including the in the present case of free dimer models driven purely by entropy. As demonstrated by the previous work of some of the present authors, the RSMI-optimal coarse-graining mappings *are* the operators with the lowest-scaling dimensions. Therefore, our procedure yields the RG relevant modes without even referring to low-energy, which is, in

fact, another proxy. We have added a discussion in the revised version to clarify this point.

Referee #3: *One could also point to the abstract, where it is claimed that “...the vastness of configurational spaces present[s] a fundamental challenge to theoretical tools, such as the renormalization group, which were so successful in characterizing the universal physical behaviour in critical phenomena.” This vastness is common to *every* example where the renormalization group is studied—every quantum field theory, and every statistical mechanics model. So, to the extent that these tools were successful, it is clearly not the fundamental challenge.*

Authors: While we of course agree with the Referee that all configuration spaces are generally vast, our intention in writing the above sentence, perhaps ineptly, was to bring the attention of the Reader that such problems have the potential to be complex, since a brute search is not an option. Some may not be, as the Referee points out, however this is not granted especially in exotic scenarios such as the one we undertook. Specifically, the absence of translational invariance makes the problem generically much more difficult. Equally, in real-space cases if the emergent DOF requires to be defined in an area containing a large amount of microscopic degrees of freedom, the difficulty of solving the appropriate optimization problem grows dramatically. We would like to draw the Referee’s attention to the complex large scale structure in the largest coarse-graining maps we have obtained, which have support on $\geq 10^3$ microscopic sites – solving this optimization problem was simply not possible numerically until the introduction of the ML methods we have used. While in pure abstract terms an optimization problem on a 3x3 block may be said to be no different to one on 100x100, in terms of numerical methods there very much is a fundamental challenge and difference, and qualitative changes in methods are required to overcome it.

This is the numerical standpoint which we had in mind when writing the introduction, and we apologise if we were not successful in formulating this in a convincing fashion. In the revised manuscript we modified the sentences in question to clarify this point.

Referee #3: *Lastly, at the end, the authors claim that we “have, in effect, reached a point where ML techniques can not only assist, but facilitate progress in theoretical physics.” I fail to see how anything in the present paper has anything to do with this claim—which should not be understood as a criticism of the work done in the paper, but rather as evidence that the claim is out of place in this article. The authors explore a proposed construction of effective degrees of freedom which is computationally unwieldy; they use a computer and a specific kind of software to do this; they produce numerical output which is interesting, and leads to a conjecture, which they then go on to prove (in the companion work [40]). How is this, operationally or qualitatively, any different than the fact that the conjecture of Birch and Swinnerton-Dyer was formulated based on numerical evidence that was produced with the help of a computer? That happened in the early 1960s. The paper’s relationship to [40] should also be articulated more clearly earlier on in the text.*

Authors: While we to a large degree understand the Referee’s point of view and accept the criticism, we would like to clarify why we wrote the sentence in the first place, and a few minor points with which we do not fully agree. Nevertheless, we decided to remove this sentence.

First, we note that the main ambition of our work is to provide practical numerical tools with which the beautiful machinery of RG may be brought to bear (after, perhaps, suitable generalizations) on large scale data-driven problems as encountered in the modern complex systems research. From this practical and numerical perspective the development of tools which are able to efficiently deal with large scale inputs is a qualitative change which was enabled by the progress in ML in the last decade. In our opinion this progress in numerical methods does not deserve to be trivialised – taking such argument to the extreme it might be said that the use of electronic computers, as in the work of Birch and Swinnerton-Dyer, did not in fact represent a qualitative change from XIXth century, when Gauss used a human computer. This is perhaps true from the point of view mathematics, but certainly not physics.

We also do not agree with the characterization of our algorithm as “unwieldy” since it uses a computer. We fully appreciate the importance of theory, but feel compelled to point out that very little in physics is rigorous by the standard of a mathematical proof, including the vast majority of the “analytical” work. Numerical methods have become crucial in theoretical physics, and a computer and algorithmic knowledge equally important tools as a good grounding in algebra. These are not in opposition – it is in fact our goal to make the application of modern numerical tools of ML as rigorous and close to physical concepts as possible. While, of course, their usage can never replace a mathematical proof, they may give researchers a much more rigorous framework and much more formally interpretable tools than some of the ad hoc solutions frequently used even in analytical works, of which the RG literature is actually full. In particular, in the future, we intend to combine our methods with symbolic computation tools to, for instance, automatically characterise the algebraic properties, such as symmetries, that the constructed coarse graining transformations may obey.

Furthermore, we would like to point out that, our numerical algorithm is fully open source, allowing any researcher to examine, test and modify the details without the use of any commercial software.

With respect to the Ref. [40] (cited as [27] in the revised manuscript): this work, by one of the authors of the present manuscript, provides an alternative, and more “analytic” formulation of the effective DOFs for the example model studied, which, crucially, yields the same physical picture, and additionally numerical works studying the correlations in more detail. It is key to note however, that the algorithm we present here is applicable to a broad class of problems (as shown by e.g. the new Appendix D), while the solution in Ref. [40] is specific to the dimer problem on the Amman-Beenker tiling, and does not generalize. Lastly, we would like to note that our result also goes beyond proving the conjecture about the *existence* of discrete scale invariance - we explicitly constructed a concrete set of collective degrees of freedom that manifest this property. These objects can now be used for different tasks, *e.g.*, a better estimation of the critical exponents.

We hope we have managed to explain our reasoning to the Referee in a convincing fashion. We do acknowledge, however, that the sentence as written may have come across as grandiose, and following the Referee’s suggestion we have decided to remove it.

Referee #3: In conclusion, I can only recommend publication of this paper after serious revision, to clarify precisely what is done in it, chart its relationship to the authors’ other related publications, to

better highlight the results and their physical and conceptual importance, and to eliminate unnecessarily florid, misleading, or unclear language wherever possible.

Authors: We thank the Referee for the detailed comments and suggestions. We have implemented the changes in the revised manuscript, and we hope that the Referee will find the revised version more clear and to the point.

Summary of Changes

1. We have added the new Appendix B, where we applied RSMI-NE to a frustrated antiferromagnets on more generic random graphs where a canonical choice of block shapes is not known a priori.
2. We have added the new Appendix D, where we show that the RSMI-NE algorithm can reproduce the conclusions regarding the discrete-scale invariance of the dimer model on AB tiling, without a fine-tuned choice of the blocks that are informed by the underlying structure.
3. We have changed the wording “constructively proving an open conjecture” in the revised version to “explicitly constructing emergent DOFs that confirm an open conjecture”.
4. We now mention in the main text and the last sentence of the abstract the new application of our method to the minimum bisection problem on random graphs.
5. We have extended the second paragraph to discuss how the proxy of wavelength helps performing RG in translationally invariant systems, but is not useful for inhomogeneous systems.
6. From the third till sixth paragraphs we added more details of our lossy-compression principle and the RSMI-NE algorithm, mentioning that our algorithm is based on the formal links between the notions of relevance in information bottleneck and RG.
7. We have added a new paragraph to outline the structure of the paper.
8. In the end of the paragraph that starts with “To uncover the emergent DOFs...”, we have added a new sentence to clarify that our method does not rely on a fine-tuned choice of the block shape.
9. In the penultimate paragraph on the third page, we have clarified that class-8 (3) patches are the green (blue) patches shown in Fig. 2.
10. In order to avoid the confusion that \mathcal{V} are not the classes of coarse grained blocks, but are random variables containing the degrees of freedom on such a block, we now refer to the subgraphs with a distinct notation V in the fourth paragraph. Also, on page 4, in the paragraph starting with “Having found the emergent DOFs...” we have made the following change: “emergent DOFs in each class \mathcal{V} ” \rightarrow “emergent DOFs in different classes of blocks \mathcal{V} ”.
11. We added a new paragraph at the end of the results section and before the conclusion to connect the terms “exotic criticality” and “broken conformal symmetry” to our results and to discuss the physical significance of these features.

Response to Reviewer #2:

⚡ I thank the authors for their response. Overall, an existing method (cf. Refs. 17–19 and 29) has been applied to a known problem with a solution that has been derived in a separate work (cf. Ref. 26). While the authors have included an additional example in the Appendix, I find the overall contribution unconvincing. Furthermore, many parts of the manuscript remain unclear. This lack of clarity is primarily due to insufficient communication of the actual contributions and presentational issues regarding the employed algorithm.

We were sorry to read that the Reviewer continued to have misgivings about our work. We address the Reviewer’s specific points, as they are raised, below. We thereby hope to address the more generally stated concerns in the previous paragraph.

However, before proceeding, we must correct a factual error in the Reviewer’s account. The Reviewer states that our method ”has been applied to a known problem with a solution that has been derived in a separate work (cf. Ref. 26)”. However, our work pre-dates Ref 26, as can be seen from the dates at which both works appeared on the arXiv: 27/1/23 for ours [arxiv.org/abs/2301.11934], compared to 15/2/23 for Ref 26 [arxiv.org/abs/2302.07879]. We submitted our work to Nature Physics on 6/2/23 [NPHYS-2023-02-00334], before Ref 26 appeared. We added a reference to Ref 26 as a courtesy, but the order of precedence is clear.

We hope this clarification changes the Reviewer’s understanding of the present work. Rather than test a known result, we actually established that result in the present paper. Certain details of our solution, specific to the problem of classical dimers on the Ammann Beenker tiling, were then elaborated upon separately by one of us in Ref 26.

⚡ (1) The contributions of this paper remain unclear. It primarily applies an existing method (cf. Refs. 17–19), but this is not clear from the revised introduction. In the first two pages, the focus is mainly on an algorithm overview, which suggests that this is a main contribution of the paper.

Our phrasing was unclear here. It is not the case that we simply apply an established method. The method developed in 17-19 could not be applied to inhomogeneous, non-periodic systems. Our advance is to develop a new approach to RSMI-NE which works for this much broader class of problems.

To draw an analogy, the 2D Ising model is solved on the square lattice, but it is far from being solved on arbitrary inhomogeneous bipartite lattices. The methods used in the periodic case simply do not apply in inhomogeneous settings. In the same way, it is not correct to say that we apply existing methods. We present major advances on those methods, both conceptual and numerical. Our new resulting algorithm can then be applied in a much broader setting.

⚡ Additionally, the authors’ explanation in their response letter that the employed numerical methods, particularly the machine-learning approaches, are subordinate is confusing. If these methods are not central, why do the authors state in the introduction that advancements in machine learning are a key part of their method without further detailed explanation?

We hope to have addressed this point in our reply to the previous point. However, since it is clear that

we did not explain this with sufficient clarity in the manuscript, we have now added a paragraph to the Introduction as follows:

Here we tackle the challenge of inhomogeneity in complex systems with vast configuration spaces. Formulating the RG of an inhomogeneous system as a lossy compression of information [12, 13] on a graph allows us to define the procedure in a *geometry-independent* manner. This key step is based on the observation that the compression theoretic RG some of us introduced for lattice systems [14-16] can be mathematically generalized to arbitrary graphs, yielding RG informed both of the interactions and of the spatial relations, thus overcoming a major conceptual challenge. The numerical execution of this *data-driven* procedure entails the difficult task of estimating mutual information for large-dimensional random variables on graphs. This is achieved by using recent advances in machine learning, formulating it as a classification task of distinguishing jointly sampled pairs of variables from those sampled independently (Appendix A), *i.e.* using contrastive learning [17-19], and extending the computational tools of Ref. [14].

ii. It would be much clearer to explain at the very beginning that this paper focuses on applying an existing method to problems whose solutions have already been discussed in prior work.

We hope that it is now clear that this is not the case. We have updated the manuscript as mentioned in the previous reply, in order to more clearly state what it is that we have done.

Since both other referees understood this point, we hope this additional clarification means that all readers can now clearly understand the advance of this work.

ii. (2) The revised method description on pages 1 and 2 is still difficult to follow. The method that the authors employ is a key part of the paper, but the reader is not properly introduced to it. One has to continuously switch between this part, other parts of the manuscript, and Refs. 17–19. For example, the authors mention subgraphs V^i without defining the actual graph. Mutual information appears in the new Eq. (2), but it is neither properly defined nor adequately explained. Instead the authors refer to their prior papers on this topic (Refs. 17–19 and 29). On page 2 the authors mention that they will not further distinguish between “the local graph” and “the configurations” because the “correct meaning is implied by the context”. The term “local graph” appears once in the whole manuscript. Is it the same as V^i ? Overall, this revised section is not clear.

We agree with the Reviewer that this section required further work. To implement the Reviewer’s suggestion we have gathered everything the reader needs, in sufficient detail, at the start, rather than referring elsewhere. In the revised introduction we now explicitly define mutual information and give its qualitative meaning. In addition, we define the graph G and explain that the V^i are local subgraphs. Moreover, we now make a clear distinction with the graph structure V^i and the degrees of freedom \mathcal{V}^i supported on the edges of the graph.

ii. (3) The main objective of the new appendix section is not well-motivated. The authors apply the previously introduced method to the Ising model on an almost bipartite graph. As described by the

authors, the problem of finding the ground state is tractable for a bipartite graph. Keeping the graph close to a bipartite one (as in the added simulation results) ensures that the problem is simple enough. The authors then demonstrate that their method can, e.g., correctly identify the partitioning and solve the minimum bisection problem, given an almost bipartite graph.

On this point we must disagree. Adding even a single non-bipartite connection to a bipartite graph entirely changes the associated mathematics and physics. For example, the dimer model on a bipartite planar regular graph admits an emergent $U(1)$ gauge theory. Adding a single edge to render the graph non-bipartite results instead in a \mathbb{Z}_2 gauge theory. The latter admits deconfined phases, while the former does not. To take another example, Lieb's theorem states that bipartite graphs admit topologically protected electronic zero modes (the number given by the imbalance between bipartite sublattices); again, adding a single connection will remove this protection. Mathematically, no result proven only for bipartite graphs can be expected to survive the addition of a single non-bipartite connection.

For this reason, the new results we present in the appendix are non-trivial.

Nevertheless, we have rewritten the Appendix B to better motivate the specific random graph ensemble we constructed. In particular, we clarify that this ensemble provides a tuning parameter to move away from perfectly planar graphs (in addition to perfect bipartiteness), for which there are known polynomial-time algorithms. Therefore we can tune the hardness of the bipartitioning problem without losing track of the correct (planted) solution. This is useful for benchmarking, which we have now added to the revised version of the manuscript.

⚡ **(a)** *A comparison with existing (exact and approximate) graph optimization approaches is missing. Based on the current results, it isn't clear if the presented method holds any computational advantage over existing graph-bisection approaches. What is the size of the considered graph and what is the computational complexity of the employed algorithm?*

We agree this is a sensible check on our results. We have therefore added the requested comparison. We thank the Reviewer for the suggestion, which we agree has improved the quality of the presentation.

We have benchmarked the bipartitioning performance of the RSMI-NE algorithm against a greedy local search algorithm and the Goemans-Williamson algorithm (which has a performance guarantee for reaching at least 87% of the correct partitioning) [Goemans and Williamson, JACM **42**, 6 (1995)]. We provide comparisons for both the efficiency and the quality of the bipartitioning.

The local greedy search algorithm is expected to perform well in the regime we considered, and it can indeed find the global optimum for a range of values $0\% < f < 7.5\%$. However, its convergence time scales poorly with the graph size (see Fig. S6), and it does not have an approximation guaranteed to approach the optimum solution in general (it can get trapped in local minima). We found that the RSMI-NE algorithm achieves comparable performance to the local greedy search at significantly smaller runtimes (Figs. S5a and S6). Furthermore, for graphs with $f < 10\%$, RSMI-NE exceeds the approximation guarantee for the Goemans-Williamson algorithm (approximation ratio 87% for generic graphs), which is a standard benchmark for graph bipartitioning algorithms (Fig. S5b).

⋮⋮ **(b)** *What would the order parameter dependence on β look like if one (i) wouldn't employ any coarse graining or (ii) combine an existing bisection approach with some kind of naive coarse-graining baseline? Such comparisons can help provide a mechanistic interpretation of \mathcal{O}_Λ in the considered example.*

We included new results to address this question. The new Figure S7 shows the temperature dependence of the order parameter constructed by the one-exchange bipartitioning algorithm, both in combination with a block-wise coarse graining (Fig.S7b, which can be compared directly with Fig.S4d) and without any coarse graining (Fig.S7a). We found that this new computational observable has almost identical behaviour to the antiferromagnetic order parameter constructed by the optimal RSMI-NE compression. Indeed this is to be expected, since RSMI-NE achieves an approximation ratio of ≈ 0.95 for the bipartitioning on the graph considered here ($f = 5\%$).

⋮⋮ **(4)** *The vertical error bars in Fig. S5 are not defined.*

We thank the Reviewer for highlighting this omission. We have provided the definition in the caption of Fig. S5: The error bars show the standard deviation of the mutual information value over 5 independent runs of the RSMI-NE optimisation.

⋮⋮ **Minor:** *Reference 21: “Nonequilibrium fixed points of coupled ising models” \rightarrow “Nonequilibrium fixed points of coupled Ising models”*

⋮⋮ *Reference to the following paper is missing: Wolff, U. (1989). Collective Monte Carlo updating for spin systems. Physical Review Letters, 62(4), 361.*

We thank the referee for pointing out these issues. We have corrected the format of Reference 21, and added the requested reference on the Wolff cluster algorithm.